

# Attribution of Atmospheric Sulfur Dioxide over the English Channel to Dimethylsulfide and Changing Ship Emissions

Mingxi Yang[1*], Thomas G. Bell[1], Frances E. Hopkins[1], Timothy J. Smyth[1]

[1] Plymouth Marine Laboratory, Prospect Place, Plymouth, UK PL1 3DH.

*Correspondence to*: M. Yang (miya@pml.ac.uk)

**Abstract.** Atmospheric sulfur dioxide ($SO_2$) was measured continuously from the Penlee Point Atmospheric Observatory

(PPAO) near Plymouth, United Kingdom between May 2014 and November 2015. This coastal site is exposed to marine air

across a wide wind sector. The predominant southwesterly winds carry relatively clean background Atlantic air. In contrast, air

from the southeast is heavily influenced by exhaust plumes from ships in the English Channel as well as near the Plymouth

Sound. New International Maritime Organization (IMO) regulation came into force in January 2015 to reduce sulfur emissions

tenfold in Sulfur Emission Control Areas such as the English Channel. Our observations suggest a three-fold reduction from

2014 to 2015 in ship-emitted $SO_2$ from that direction. Apparent fuel sulfur content calculated from coincidental $SO_2$ and carbon

dioxide ($CO_2$) peaks from local ship plumes show a high level of compliance to the IMO regulation (>95%) in both years.

Dimethylsulfide (DMS) is an important source of atmospheric $SO_2$ even in this semi-polluted region. The relative contribution

of DMS oxidation to the $SO_2$ burden over the English Channel increased from ~1/3 in 2014 to ~1/2 in 2015 due to the reduction

in ship sulfur emissions. Our diel analysis suggests that $SO_2$ is removed from the marine atmospheric boundary layer in about

half a day, with dry deposition to the ocean accounting for a quarter of the total loss.

## 1 Introduction

The trace gas sulfur dioxide ($SO_2$) is important for atmospheric chemistry (Charlson and Rodhe, 1982) as a principal air pollutant

(e.g. contributor to acid rain). Atmospheric oxidation of $SO_2$ leads to sulfate aerosols, which influence the Earth's radiative

balance directly by scattering incoming radiation and indirectly by affecting cloud formation (Charlson et al. 1987). Important

natural sources of $SO_2$ include the atmospheric oxidation of dimethylsulfide (DMS, which is formed by marine biota) and

volcanic eruptions. Anthropogenic fossil fuel combustion also produces $SO_2$. $SO_2$ is removed from the lower atmosphere by dry

deposition and oxidation in both the gas phase and the aqueous phase. The relatively slow gas phase oxidation of $SO_2$ leads to

sulfuric acid vapor, which usually condenses upon pre-existing aerosols but can nucleate to form new particles under specific conditions (e.g. Clarke et al. 1998). The much faster aqueous phase oxidation of $SO_2$ takes place primarily in cloud water (e.g. Hegg 1985; Yang et al. 2011) and leads to particulate sulfate, which is subsequently removed from the atmosphere by wet deposition.

$SO_2$ production from DMS occurs mainly via daytime oxidation by the hydroxyl radical (OH). From observations in the Equatorial Pacific, Bandy et al. (1996) and Chen et al. (2000) reported a clear increase in $SO_2$ mixing ratio and coincidental decrease in DMS during the day. This anti-correlation confirmed that DMS oxidation by OH is an important source of $SO_2$ over the remote ocean. More recently, Yang et al (2011) showed that DMS remains the predominant sulfur precursor in the marine boundary layer of the relatively unpolluted Southeast Pacific.

In recent decades, $SO_2$ emissions from terrestrial combustion sources such as power plants and ground transportation have been subject to strict regulation (e.g. UK Clean Air Acts). These forms of legislation have significantly reduced the atmospheric sulfur burden over land in North America and Europe (e.g. Lynch et al. 2000; Malm et al, 2002; Vestreng et al, 2007). Unlike terrestrial $SO_2$ emissions, ship emissions were for a long period excluded from international environmental agreements. This allowed ships to burn low-grade fuels with high sulfur content (i.e. heavy fuel oils), which resulted in large

$SO_2$ emissions from ship engine exhausts (hereafter ship emissions). In addition to $SO_2$, ship exhausts also contain carbon dioxide ($CO_2$), nitrogen oxides, carbon monoxide, heavy metals, organic toxins, and particulates (e.g. Agrawal et al 2008). Closer to the coast and near shipping lanes, ship emissions can be an important source to the atmospheric sulfur budget (Capaldo et al. 1999; Dalsoren et al, 2009). Eyring et al. (2005a) estimated that global, transport-related emissions of $SO_2$ from ships in the year 2000 were approximately 3-fold greater than from road traffic and aviation combined. Air quality models predict that

aerosols resulting from ship emissions contribute to tens of thousands of premature mortality near coastlines (Corbett et al., 2007). Impacts may be further exacerbated as the global population expands and shipping-based trade increases (Eyring et al., 2005b).

In January 2015, new air quality regulations from the International Maritime Organisation (IMO), an agency of the United Nations, came into force. These regulations aim to reduce sulfur emissions tenfold in Sulfur Emission Control Areas

(SECAs) from 1% (regulation since 2010) to 0.1% by mass. The English Channel and the surrounding European coastal waters are within a SECA. The IMO has also set a target for open ocean emissions of $SO_2$ to be reduced from the current level of 3.5% to 0.35% by 2020. Winebrake et al. (2009) estimated that such a reduction in sulfur content would approximately halve the premature mortality rate in coastal regions. A decrease in anthropogenic sulfur emissions is also expected to make DMS a relatively more important sulfur source in regions such as the North Atlantic.



There have been few direct measurements of ship emissions that are relevant for a regional scale. Kattner et al. (2015) reported large reductions of $SO_2$ in ship plumes from 2014 to 2015 near the mouth of the Hamburg harbor on the river Elbe, which is about 100 km away from the North Sea. Based on the $SO_2$:$CO_2$ ratio in ship plumes, they found a high compliance rate of ~95% after the stricter regulation in January 2015. Based on measurements from the Saint Petersburg Dam, which spans the

Gulf of Finland and separates the Neva Bay from the rest of the Baltic Sea, Beecken et al (2015) found a compliance rate of 90-97% in 2011 and 2012. Compliance checks are mainly limited to manual checking of fuel logs and fuel quality certificates when ships are in port.

      Global and regional ship emission estimates have typically been scaled from inventories for individual vessels in combination with information about ship traffic (e.g. Endresen et al. 2003; Collins et al. 2008; Matthias et al. 2010; Whall et al.

2010; Aulinger et al. 2015; Jalkanen et al. 2016). Accurate assessment of the success of IMO regulations requires long-term, continuous observations at strategic locations. Here we present 1.5 years of continuous atmospheric $SO_2$ measurements from the Penlee Point Atmospheric Observatory (PPAO) in the English Channel, one of the busiest shipping lanes in the world. The PPAO measurements date back to seven months before recent IMO sulfur regulations came into force, providing a reference point for future changes in emissions. The unique location of PPAO (Figure 1;

http://www.westernchannelobservatory.org.uk/penlee/) allows us to partition the atmospheric $SO_2$ budget to natural (mostly dimethylsulfide) and anthropogenic (mostly ship emission) sources.

## 2 Experimental

The Penlee Point Atmospheric Observatory (PPAO) is located on the western side of the mouth of the Plymouth Sound. See

Yang et al (2016) for detailed site description, a picture of the observatory, and a map of the local coastline. About 11 m above mean sea level and ~30 m away from the high water mark, the site is exposed to air that has travelled over water across a wide wind sector (from northeast to southwest). Near-continuous measurements of $SO_2$, $CO_2$, ozone ($O_3$), methane ($CH_4$), as well as standard meteorological parameters have been made at the PPAO since May 2014.

      $SO_2$ mixing ratio was measured every 10 seconds by a Trace Level-Enhanced Pulsed Fluorescence $SO_2$ analyzer

(Thermo Scientific, Model 43i). The instrument noise level is about 0.06 parts per billion (ppb) at a one-minute averaging interval and 0.025 ppb at a five-minute averaging interval. $O_3$ mixing ratio was measured every 10 seconds by a dual-beam, UV absorption Ozone monitor (2B Technologies, Inc. Model 205), which has a noise level of about 2.2 ppb at a one-minute averaging interval and 1.0 ppb at a five-minute averaging interval. The $SO_2$ and $O_3$ sensors shared the same air inlet, which consisted of a ~4 m long 0.64 cm outer diameter PFA (perfluoroalkoxy) tubing that extended just outside the air vent of the

building (~2 m above ground). A 5 micron diameter Teflon filter was installed upstream of the instruments to reduce the



particulates in sampled air. The $O_3$ instrument sampled through an additional 5 micron particle filter. These filters were replaced approximately every 2-4 weeks. $SO_2$ and $O_3$ were blanked simultaneously by directing sample air through the existing particle filters, ~8 m of 0.64 cm outer diameter copper tubing (which efficiently removed $SO_2$), and an $O_3$ scrubber. Linearly interpolated blanks are subtracted from the raw data. We checked the calibration of the $SO_2$ instrument occasionally with a $SO_2$

gas standard diluted in nitrogen (100 ppb, BOC). The measured $SO_2$ mixing ratio was within a few percent of the gas standard. An $O_3$ calibration device was unavailable during the duration of these measurements. However, an intercomparison with a recently-calibrated 2B $O_3$ Monitor showed that the accuracy of the $O_3$ measurement at PPAO was within ±5%.

CO_2$ mixing ratio was measured by a Picarro cavity-ringdown analyzer (G2311-f) every 0.1 second (See Yang et al. 2016 for details). Ambient air was drawn from a mast on the rooftop of the observatory through a ~18 m long 0.95 cm outer

diameter perfluoroalkoxy (PFA) tubing at about 15–30 L min$^{-1}$ by a dry vacuum pump. The Picarro analyzer subsampled from this main flow via a ~2 m long 0.64 cm outer diameter Teflon PFA tubing and a high throughput dryer (Nafion PD-200T-24M) at a flow rate of ~5 L min$^{-1}$. The total delay time from the inlet tip to the analyzer was 1.9–3.3 seconds. The instrument calibration was checked with a $CO_2$ gas standard (BOC) and the accuracy was within 0.5%. The instrument noise for one minute-averaged $CO_2$ is less than 0.05 ppm.

SO_2$, $O_3$, and meteorological parameters data were logged and timestamped by the same PC. Given the short inlet tubing length and the specified flow rates, the calculated delay time for these gases from the inlet tip to the sensors was less than four seconds. $CO_2$ data were logged on a separate computer (Picarro internal PC). Both PCs were synchronized to network UTC clocks once a week. The time difference between the $SO_2$/$O_3$/meteorology measurements and the $CO_2$ measurements was less than a minute.


### 3 Results

Figure 1 shows the frequency distribution of winds from 2014 to 2015 as a wind rose (distributions were nearly identical for these two years). Winds predominantly came from west/southwest at moderate-to-high speeds, which were typically associated with low pressure systems in the North Atlantic and generally carried low mixing ratios of $SO_2$ and $CO_2$ (Yang et al. 2016). The

wind sector between 110 and 250° is completely unobstructed by land. The Rame Head peninsula sits ~1.5 km west of PPAO, beyond which lie ~40 km of coastal seas, the county of Cornwall (width of ~30 km), and the North Atlantic Ocean. Winds from between 50 and 110° face the eastern side of the Plymouth Sound, which is busy with ship traffic. The southeasterly sector is likely affected by emissions from local ships as well as distant pollution from the English Channel and continental Europe. Commercial ferries enter the Plymouth Sound at least once a day from approximately due south.



Atmospheric $SO_2$ and humidity varied significantly at PPAO depending on wind direction (Figure 2). Increased relative humidity clearly indicates the marine-influenced wind sector between ~60 and 260°. $SO_2$ mixing ratios were higher and more variable when the air mass had travelled from the southeast than when it had come from the southwest. This elevated $SO_2$ signal was more pronounced in 2014 (averaged between May and December) than in 2015 (averaged between January and November).

The lowest $SO_2$ mixing ratios were observed in the western, terrestrially-influenced wind sector in both years. In the Appendix, we show episodes of large $SO_2$ plumes from the Icelandic volcano Bardarbunga as observed at PPAO. These volcanic events do not affect our analysis of $SO_2$ in the marine atmosphere since winds were from the northwest.

### 3.1 Ship Plumes and $SO_2$ Frequency Distributions

Figure 3 shows a ship plume-influenced time series during a period of southeasterly winds. Sharp spikes in $SO_2$ coincided with spikes in $CO_2$ (e.g. at about 12:00 and 15:40 UTC). $O_3$ was significantly depleted in plumes because of its reaction with nitrogen oxides (NOx) emitted from ships. These plumes tended to last just a few minutes and likely corresponded to local (within a few km) ship emissions. A lower, broader peak in $SO_2$ can also be observed between about 18:30 to 20:00 UTC. This was likely due to more distant ship emissions that have been diluted and mixed in the atmosphere. In Section 4.2, we use the ratio between the

$SO_2$ and $CO_2$ peaks to estimate the ship's apparent fuel sulfur content.

Figure 4 shows the histograms of $SO_2$ mixing ratios (from 5-minute averages) in 2014 and 2015. We have separated the data to two wind sectors, southeast (80-170°) and southwest (210-250°). The southeast sector encompasses the English Channel while avoiding most of the UK landmass. The southwest sector, with largely an unobstructed oceanic fetch of thousands of kilometers, bypasses the northwestern coast of France as well as the busiest part of the shipping lanes. In both sectors the $SO_2$

distributions shifted towards lower mixing ratios in 2015 compared to 2014. For example, in the southeast sector $SO_2$ mixing ratios above 0.5 ppb occurred less frequently in 2015 (~1%) compared to 2014 (~11%).

### 3.2 Diel Variability in $SO_2$

We compute the mean diel cycles of $SO_2$ mixing ratio in the southeast and the southwest sectors for both 2014 (May to

December) and 2015 (January to November), which are shown in Figure 5. The long averaging periods help to reduce measurement noise and also allow variability caused by horizontal transport to largely cancel. $SO_2$ from the southwest shows a very tight diel cycle and low variability (relative standard errors less than 10%). $SO_2$ mixing ratio was the lowest near sunrise, increased throughout the day, and decreased after sunset. This diel cycle suggests that $SO_2$ from the southwest came primarily from the photooxidation of biologically-derived DMS. Differences in the mean $SO_2$ diel cycles in 2014 and 2015 for the



southwest sector are largely due to the different months used in the averaging. Considering only the months of May to November, mean $SO_2$ mixing ratios in this wind sector for the two years differ by only ~0.01 ppb.

$SO_2$ from the southeast was about three times higher and also more variable than from the southwest in 2014. Peaks in $SO_2$ were observed in the morning, mid-afternoon, and early evening, consistent with the schedule of a commercial ferry service.

In 2015, $SO_2$ from the southeast was about two times higher than from the southwest and variability in $SO_2$ mixing ratio was reduced. In both years, $SO_2$ from the southeast shows an underlying diel trend (i.e. increasing during the day and decreasing at night) that suggests contributions from DMS oxidation. This implies that $SO_2$ from the southeast is made up of at least two major components: ship emissions and DMS oxidation.

### 3.3 $SO_2$ from DMS Oxidation

The $SO_2$ diel cycle from the southwest wind sector (Figure 5) is consistent with daytime $SO_2$ production from DMS oxidation by the OH radical (e.g. Bandy et al. 1996; Yang et al. 2009). DMS was measured using a high resolution proton-transfer-reaction mass spectrometer from the rooftop of the PML building in Plymouth (~6 km north/northeast of PPAO) in 2012 (Yang et al. 2013) and in 2015. Recently-calibrated transmission efficiencies from the manufacturer (Ionicon, Austria) and kinetic reaction

rates from Zhao and Zhang (2004) were used to derive the DMS mixing ratio. DMS levels in marine sector air from the southwest and southeast were comparable during the 2015 measurement period (21 April to 15 May). The mean DMS diel cycle from the marine sector (Figure 6) clearly shows an anti-correlation with shortwave irradiance, implying daytime oxidation by OH (mostly to $SO_2$). The diel amplitude in DMS mixing ratio was ~0.09 ppb. A day/night difference in DMS was also observed from the PML rooftop in June 2012 from the marine wind sector (Yang et al. 2013).

The conversion efficiency from DMS to $SO_2$ due to OH oxidation is about 70-90% (Davis et al., 1999; Chen et al., 2000; Shon et al., 2001). At a conversion efficiency of 80%, 0.09 ppb of oxidized DMS would lead to about 0.07 ppb of $SO_2$ produced during the day. In comparison, over the 1.5 years of observations at PPAO the mean amplitude of the $SO_2$ diel cycle from the southwest was ~0.06 ppb. This comparison is qualitative because DMS was only measured during the spring/summer periods and at a different location. Nevertheless, it appears that DMS oxidation can approximately account for the observed $SO_2$

diel cycle from the southwest wind sector.

### 3.4 Removal of $SO_2$ from the Marine Atmosphere

$SO_2$ is mainly removed from the marine boundary layer via aqueous oxidation (e.g. cloud processing) and deposition to the ocean. We can approximate the total loss of $SO_2$ from the nighttime change in the averaged $SO_2$ mixing ratio (daytime $SO_2$

oxidation is very slow). This calculation assumes no temporal trend in any nocturnal ship emissions (on a diel timescale) and a

constant marine boundary layer height. In a polluted marine environment, a small amount of $SO_2$ could be formed at night via DMS oxidation by the nitrate radical ($NO_3$; Yvon et al. 1996), a process we neglect here. A linear fit to the nighttime decrease of $SO_2$ from the southwest in 2014 (Figure 5A) yields a total loss rate of about -0.2 ppb per day. The average $SO_2$ mixing ratio was about 0.1 ppb in the evening hours, which implies a $SO_2$ residence time of approximately 0.5 d. In 2015, the total loss rate was

about -0.1 ppb per day, which also implies a $SO_2$ residence time of ~0.5 d. This residence time is in close agreement with previous estimates in the marine atmosphere (e.g. Cuong et al. 1975; Yang et al 2011).

We compute the dry deposition flux of $SO_2$ to the surface ocean as $-V_d * [SO_2]$, where $[SO_2]$ is the atmospheric $SO_2$ concentration and $V_d$ is the deposition velocity of $SO_2$. Upon contact with seawater, $SO_2$ is rapidly oxidized to sulfate, resulting in a dissolved $SO_2$ concentration near zero in the surface ocean (e.g. Liss and Slater, 1974). Thus it is reasonable to assume that

the air-sea concentration difference of $SO_2$ resides entirely on the airside. We compute the deposition velocity of $SO_2$ using the COARE gas transfer model (Fairall et al. 2011), which uses the airside diffusivity of $SO_2$ from Johnson (2010) and the measured wind speed at PPAO. Diel cycles in $SO_2$ deposition flux (Figure 7) resemble the mirror image of $SO_2$ mixing ratio. For the southwest wind sector, the average $SO_2$ deposition flux was about -1 to -2 $\mu$mole m$^{-2}$ d$^{-1}$.

For the southeast sector, deposition flux averaged about -3 $\mu$mole m$^{-2}$ d$^{-1}$ in 2014 and -1 $\mu$mole m$^{-2}$ d$^{-1}$ in 2015.

Interestingly, while $SO_2$ mixing ratio from the southeast was still higher than from the southwest in 2015, deposition fluxes between the two wind sectors were comparable. This is because wind speeds were typically lower from the southeast. Overall, dry deposition removes $SO_2$ from a well-mixed, 1 km deep marine atmospheric boundary layer with a timescale of ~2 days at PPAO. It accounted for approximately a quarter of the total $SO_2$ losses, similar to previous findings (e.g. Yang et al. 2011).

**4 Discussion**

**4.1 Long Term Changes in $SO_2$ and Ship Emissions**

We evaluate the long-term trends in $SO_2$ from different wind sectors in order to understand seasonal variability and assess any changes due to ship emissions. $SO_2$ mixing ratios from the southeast and the southwest are averaged into monthly intervals (Figure 8). Mean $SO_2$ from the southwest in the summer months were comparable between 2014 and 2015. In contrast, $SO_2$

from the southeast was significantly lower in summer 2015 than in summer 2014. For both wind sectors, lower $SO_2$ mixing ratios were observed in winter/early spring compared to summer/early autumn. This seasonal cycle in $SO_2$ is qualitatively consistent with the annual trend in biological activity and seawater DMS concentration at the nearby L4 mooring station (Archer et al. 2009). Lower incoming irradiance and shorter daylight hours in winter will also reduce the photochemical production of $SO_2$ from atmospheric DMS. We suggest that the seasonal cycle of $SO_2$ in air from the southwest can largely be explained by

natural variability.





The difference in $SO_2$ mixing ratio between the southeast and southwest sectors ($\Delta SO_2$) is shown in Figure 9. There is a fair amount of scatter in $\Delta SO_2$, which is partly because $SO_2$ measurements from the two wind sectors were not concurrent (i.e. winds could not be blowing from the southeast and southwest at any single moment). Nevertheless, we see that mean (±standard error) $\Delta SO_2$ decreased from ~0.15 (±0.03) ppb in 2014 to ~0.05 (±0.01) ppb in 2015, with a sharp drop off coincident with the 1[st]

January 2015 mandate for ship emission reduction.

We attribute $\Delta SO_2$ to ship sulfur emissions based on the following assumptions: I) $SO_2$ from the southwest is from DMS oxidation only; II) $SO_2$ from the southeast is affected by both DMS oxidation and ship emissions; III) DMS oxidation contributes equally to the $SO_2$ burden in both the southeast and southwest wind sectors. Data from the southwest sector will almost certainly include some ship contributions but the tightly constrained diel cycle in $SO_2$ suggests this influence is fairly small. Automatic

Identification System maps (http://www.marinetraffic.com/) indicate lower ship density and greater distances between the shipping lanes and PPAO (thus lower $SO_2$ emissions, increased plume dilution and greater $SO_2$ removal) in the southwest sector than in the southeast sector. This information corroborates the idea that ship emissions from the southwest only have a minor effect on our observations.

Entrainment from the free troposphere could bring anthropogenic $SO_2$ into the marine boundary layer (e.g. Simpson et

al 2014), which is not accounted for here. We further assume negligible influence of terrestrial $SO_2$ emissions (e.g. from Continental Europe) in the southeast sector because of atmospheric dilution and rapid removal of $SO_2$ from the lower atmosphere (residence time ~0.5 d, see Section 3.4). The English Channel near Plymouth has a width of approximately 200 km. At a speed of 5 m s[-1], southeasterly winds blow over the channel in approximately half a day, comparable to the removal time.

**4.2 Local Ship Plumes and Fuel sulfur content**

The $SO_2$ signals from the southeast include local ship emissions (e.g. from ships entering/exiting the Plymouth Sound) as well as more distant emissions from the English Channel. Local emissions usually appear as sharp spikes, while more distant emissions tend to have plumes that are broader and less intense due to atmospheric dilution (e.g. Figure 3, between about 18:30 to 20:00 UTC). We use concurrent peaks in $SO_2$ and $CO_2$ to estimate the ships' apparent fuel sulfur content (FSC). The FSC calculation

assumes that all of the carbon and sulfur in fuel are released into the atmosphere during combustion. We use the word 'apparent' here because our calculation reflects the downstream emissions rather than the actual fuel composition. Ships that 'scrub' sulfur from stack emissions will have apparent FSC values that are lower than the actual fuel sulfur content. To minimize the uncertainty in our estimate, we focus on well-resolved plumes from nearby ships only.

von Glasow et al (2003) modeled the horizontal dispersion of a ship plume using:

$H = H_0 \cdot (t/t_0)^{0.75}$                    (1)





$H_0$ and $H$ are the horizontal extents of a plume at the initial time $t_0$ and the time of interest $t$. The initial plume extent ($H_0$) is assumed to be 10 m. We estimate the horizontal dispersion of plumes emitted from an upwind distance of 2 km (e.g. half away across Plymouth Sound) and 100 km (e.g. half away across the English Channel). At a speed of 8 m s$^{-1}$, wind travels 2 km and 100 km in ~4 and 200 minutes, respectively. Applying these timescales to Eq. 1 yields horizontal plume extents of 600 m and

11000 m. For a ship 2 km away that is traversing perpendicular to the mean wind at a speed of 4 m s$^{-1}$, its emission should be observable at PPAO for 2.5 minutes. A ship 100 km away would have a plume that is observable for nearly an hour. These timescales are roughly consistent with our time series observations (e.g. Figure 3). Ships that do not travel perpendicular to the wind will have plumes that are observable for longer periods. Faster wind speeds or ship speeds shorten the duration of plume detection, and vice versa. Overall, the above calculations suggest that local ship plumes have a typical duration of a few

minutes.

The following steps were used to separate local ship emissions from the background or more distant emissions based on 1 minute average gas mixing ratios. All data processing was done with Igor Pro (WaveMetrics).

1) Apply 2 hour running-median smoothing to the $SO_2$ time series

2) Identify 'no plume' times as when the 1 minute resolution $SO_2$ was within 0.1 ppb (i.e. < twice the instrument noise) of

the smoothed $SO_2$

3) Subtract the linear interpolation of the 'no plume' $SO_2$ time series from the 1 minute resolution $SO_2$ time series to derive the $SO_2$ deviations from the background ($SO_2$')

4) Apply the 'FindPeak' function to $SO_2$' to identify the time, height, leading edge, and trailing edge of a peak within non-overlapping 10-minute windows. A minimum peak height of 0.2 ppb (i.e. > three times the instrument noise) was

required for positive peak identification.

5) Integrate $SO_2$' between the leading edge and the trailing edge to yield the $SO_2$ plume peak area

The 10-minute window in step 4 was chosen to minimize the occurrence of multiple plumes within a single window but also allow for identifications of plumes persisting for several minutes. A total of 816 distinct $SO_2$ plumes were identified from May 2014 to November 2015 from the marine wind sector. The mean $SO_2$ plume height (at 1 minute resolution) was 1.35 ppb in

2014 and 0.48 ppb in 2015, with a typical plume duration of ~3 minutes.

$CO_2$ plumes from local ship emissions were identified in an analogous fashion based on an analysis of the 1 minute resolution $CO_2$ time series (i.e. independent of the $SO_2$ analysis). The 'no plume' $CO_2$ threshold (Step 2 above) was set to be 0.2 ppm, and the minimum peak height required for positive peak identification (Step 4) was set to be 1.0 ppm. Based on these schemes, a total number of 1242 separate $CO_2$ plumes were identified from May 2014 to November 2015, with a mean plume

height of 2.6 ppm.



The ships' apparent fuel sulfur content (FSC) is computed from coincidental $SO_2$ and $CO_2$ plume heights as well as plume areas (N = 245) following Kattner et al. (2015). To account for clock drift between the PCs that recorded the $SO_2$ and $CO_2$ data, we allowed the times of the $SO_2$ and $CO_2$ peaks to differ by up to one minute. The results of these calculations for the marine wind sector are shown in Figure 10. Gaps in observations were largely due to the $CO_2$ analyzer malfunctioning or winds

out of sector. The peak area and peak height methods yielded similar results. FSC from peak area appears to be slightly more variable, possibly due to a greater sensitivity toward the definition of plume baseline. Based on peak height, in 2014 the mean (± standard error) FSC was 0.17 (±0.03) %, with ~1% of the plumes exceeding the IMO threshold of 1% FSC. About 70% of the plumes were already below 0.1% FSC in 2014. In 2015, mean FSC decreased to 0.047 (±0.003) %, with ~1% of the plumes exceeding the new IMO threshold of 0.1%. FSC estimated from the peak area method shows slightly lower levels of compliance

(~95% for both years). The reduction in mean FSC from 2014 to 2015 is proportionally comparable to the decrease in $\Delta SO_2$ computed in Section 4.1.

Our FSC estimates illustrate a qualitatively decreasing trend similar to that observed near the Hamburg harbor (Kattner et al., 2015). However, our observations are different in several aspects. The vast majority of the ships entering/leaving Plymouth Sound are naval, commercial ferries, and private vessels according to the Devonport Naval Base Ship Movement

Report. The number of large container ships entering the Plymouth Sound is proportionally much lower than near Hamburg. Because the distances between ships and PPAO were not fixed (as opposed to spatially restricted sampling locations in Kattner et al. 2015 and Beecken et al. 2015), plumes observed at our site were usually more dilute and variable in duration. This variability made plume identification more challenging. We expect the bulk of the uncertainty in FSC to be due to instrument noise in $SO_2$, which should be within 20% for plumes in 2015 and better in 2014. Total uncertainty in FSC may be higher though due to

inexact plume baseline quantification and uncertainty in the threshold required for $SO_2$ spike detection. A higher $SO_2$' threshold could bias our FSC estimates towards plumes with greater sulfur content, while a lower $SO_2$' threshold would be too close to the noise level of the measurement. Long-term records of another tracer (e.g. nitrogen oxides or particle number) would allow for a more independent identification of ship plumes for the calculation of FSC. Finally, we reiterate that our FSC estimates are for well-resolved peaks from local ship plumes, which do not necessarily reflect ship emissions from the main shipping lanes of the

English Channel.

### 4.3 Top-down Estimates of $SO_2$ Emissions from the English Channel

We use $\Delta SO_2$ with local ship emissions excluded (Figure 9) to estimate distant anthropogenic $SO_2$ emissions (e.g. from the English Channel). Without local spikes (i.e. the 'no plume' $SO_2$ time series in Section 4.2), mean $SO_2$ mixing ratio from the

southwest was only slightly lower, as might be expected due to the relatively low ship density in that direction; in contrast, $SO_2$




mixing ratios from the southeast were reduced by an average of ~0.04 ppb in 2014 and ~0.01 ppb in 2015. Approximately a quarter of the $SO_2$ attributed to ship emissions in Section 4.1 ($\Delta SO_2$) was due to local ship plumes. $\Delta SO_2$ excluding local ship emissions was ~0.11 (±0.03) ppb in 2014 and ~0.04 (±0.01) ppb in 2015. Interestingly, the largest differences in $\Delta SO_2$ with and without nearby plumes occurred in the summer for both years.

We make an order of magnitude estimate for the ship emissions in the English Channel required to sustain observed $SO_2$ mixing ratios. This calculation assumes a $SO_2$ residence time of 0.5 d in a well-mixed, 1 km deep marine atmospheric boundary layer. For this scenario it would take ~9 $\mu$mole m$^{-2}$ d$^{-1}$ of $SO_2$ from ships to account for a $\Delta SO_2$ of 0.11 ppb, and ~3 $\mu$mole m$^{-2}$ d$^{-1}$ of $SO_2$ emission from ships to account for a $\Delta SO_2$ of 0.04 ppb. Simplistic extrapolation of these fluxes to the area of the English Channel (about 75000 km$^2$) yields a total ship $SO_2$ emission of ~16 Gg per year in 2014 and ~5 Gg per year in 2015.

Compared to previous inventory-based $SO_2$ emission estimates for the English Channel, our 2014 value is about 40% of that reported by Jalkanen et al. (2016) for 2011, and about a quarter of the Ship Emissions Inventory used in the UK Department for Environment Food and Rural Affairs (DEFRA) report (Whall et al. 2010).

Our $SO_2$ emission estimate is low in part because we have purposely excluded local ship plumes. Also, $\Delta SO_2$ could be an underestimate of ship emissions due to the assumption of zero ship influence in the southwest wind sector. More accurate

constraints of sulfur emissions from the English Channel from point measurements such as at PPAO probably requires modeling of air trajectory/dispersion, detailed information about ship traffic, and quantification of the source area for the $SO_2$ mixing ratio (i.e. the concentration footprint; Wilson and Swaters, 1991; Schmid 1994). A more complete description of the sulfur budget in this environment would also require sulfate concentration measurements in aerosols as well as in precipitation droplets.

**5 Conclusions**

In this paper, we analyzed 1.5 years of continuous atmospheric $SO_2$ measurements by the coast of the English Channel. $SO_2$ mixing ratio in southwesterly winds was generally low and showed a diel cycle that is largely consistent with DMS oxidation. In contrast, $SO_2$ mixing ratio was elevated and more variable in southeasterly winds due to additional contribution from ships. This ship contribution was reduced by approximately three-fold from 2014 to 2015 in response to the International Maritime

Organization regulation on sulfur emissions in European coastal waters. Apparent fuel sulfur content calculated from coincidental $SO_2$ and $CO_2$ peaks from local ship plumes suggest a high level of compliance (>95%) in both years. As ship sulfur emissions reduce, DMS becomes relatively more important to the atmospheric $SO_2$ burden, accounting for about half of the atmospheric $SO_2$ over the English Channel in 2015. Our data show that the residence time of $SO_2$ in the marine atmosphere is approximately 0.5 d, with dry deposition explaining about a quarter of the total $SO_2$ sink.




**Appendix: SO$_2$ from Volcanic Eruption**

The Icelandic volcano Bardarbunga was active for the spring and summer months of 2014. Under favorable meteorological

conditions, SO$_2$ emitted from the volcano was transported in the upper atmosphere from Iceland to the UK, where it was

entrained into the atmospheric boundary layer. SO$_2$ from volcanic eruption, with surface mixing ratios up to 30 ppb, was clearly

detected at PPAO between 21 and 24 September 2014 (Figure A1). These SO$_2$ plumes were observed all over the UK (http://uk-

air.defra.gov.uk) and were also apparent from space (e.g. O$_3$ Monitoring Instrument, http://sacs.aeronomie.be/nrt/). Winds were

generally from the northwest during these days of relatively high atmospheric pressure. Elevated O$_3$ mixing ratios and reduced

humidity generally coincided with the high SO$_2$ mixing ratios in these plumes, qualitatively consistent with entrainment from the

troposphere. These volcanic plumes of SO$_2$ were outside of our southeast and southwest marine wind sectors.

**Acknowledgement**

Trinity House (http://www.trinityhouse.co.uk/) owns the Penlee site and has kindly agreed to lease the building to PML so that

the instruments can be protected from the elements. We are able to access the site thanks to the cooperation of Mount Edgcumbe

Estate (http://www.mountedgcumbe.gov.uk/). Thanks S. Atkinson and M. Sillett (Plymouth University) for routine maintenance

of site, S. Ussher (Plymouth University), P. Agnew, N. Savage, and L. Neal (UK Met Office) for valuable scientific discussions,

and B. Carlton (Plymouth Marine Laboratory) for setting up data communication. T. Bell and M. Yang dedicate this publication

to the late Roland von Glasow, who was a source of great inspiration and support to us both.

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





**Figures**

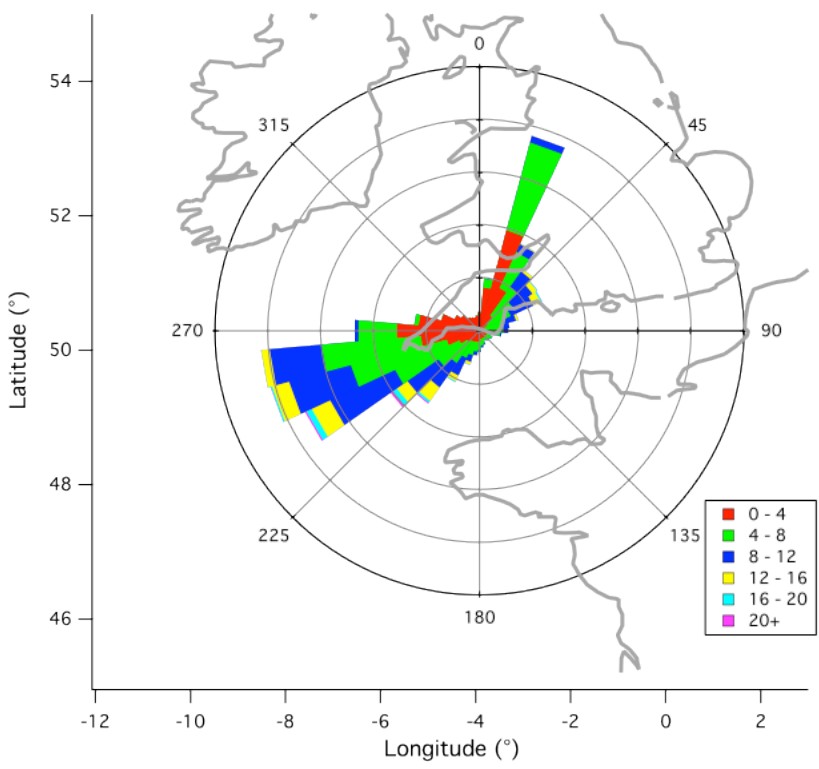

Figure 1. Wind rose at Penlee Point Atmospheric Observatory from 2014 to 2015 overlaid on a map of the British Isles. The English Channel lies between the UK and the north of France. Colors on the spokes correspond to wind speeds in units of m s$^{-1}$ and concentric circles indicate frequency of occurrence in 2.5% intervals (outer circle = 12.5%). Winds predominantly came from the west/southwest at speeds of 4-12 m/s.

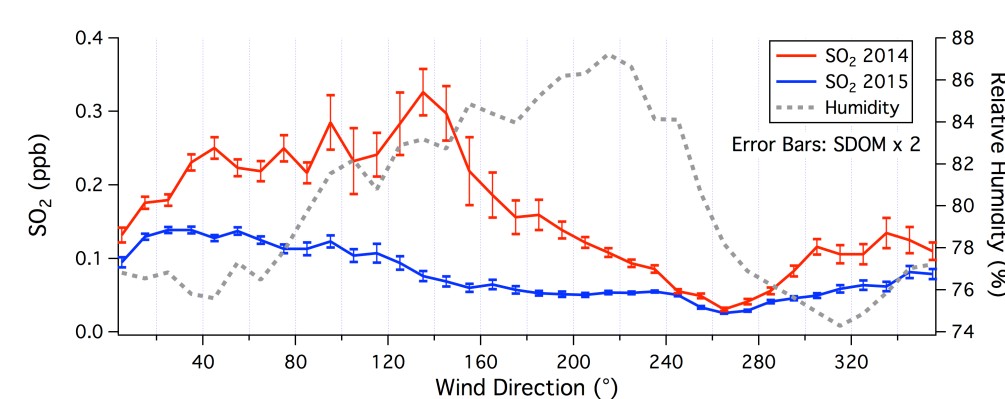

Figure 2. Averaged SO$_2$ mixing ratio and relative humidity vs wind direction for year 2014 and 2015. Error bars on SO$_2$ indicate two standard errors. Elevated humidity marks the marine-influenced wind sector to be between about 60 and 260°. Higher and more variable SO$_2$ mixing ratios were observed from the southeast direction, particularly in 2014. Icelandic volcano plumes (e.g. Figure A1) were excluded from averaging.



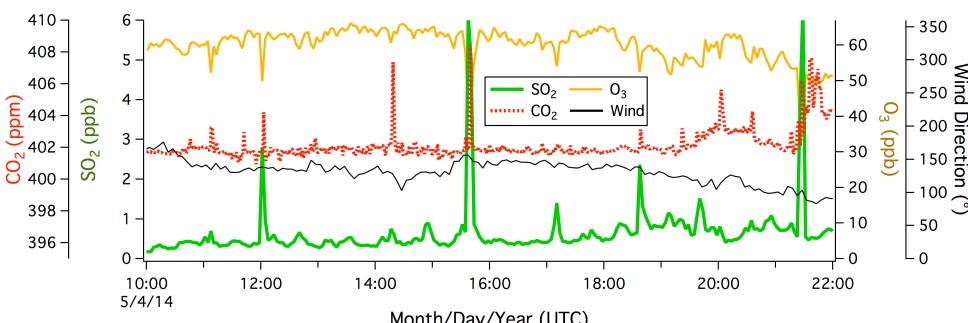

Figure 3. Example of local ship plumes on a day of southeasterly winds. Sharp peaks in $SO_2$ and $CO_2$ generally coincided with sudden depletions in $O_3$.

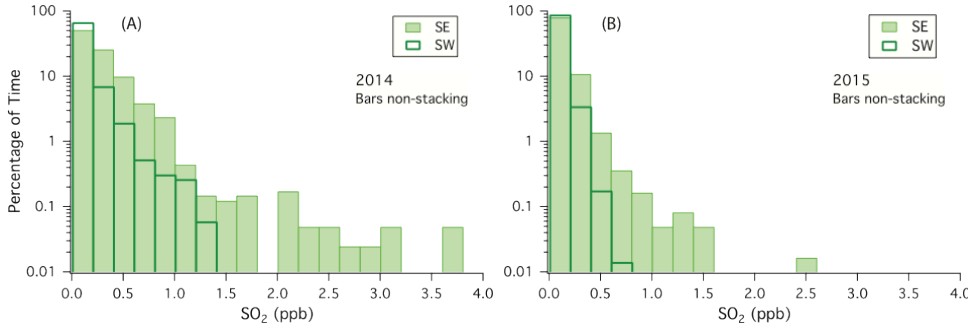

Figure 4. Histogram distributions of $SO_2$ mixing ratios in 2014 (A) and 2015 (B) from the southeast and southwest wind sectors (5-minute average). Distributions are normalized to the total number of observations from the respective wind sectors.

20





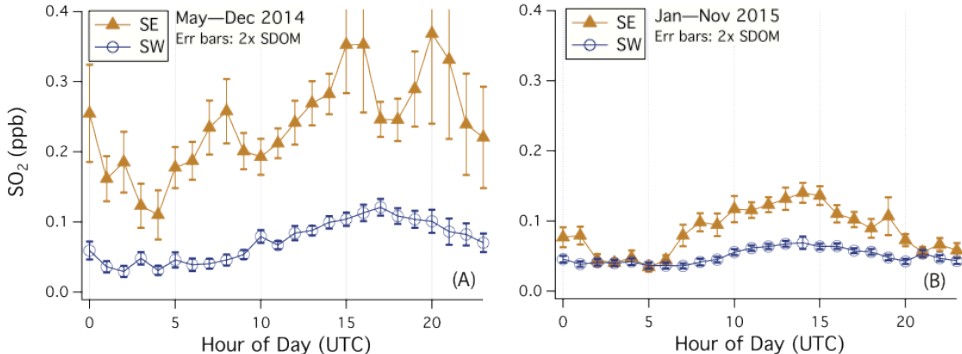

Figure 5. Average diel cycles of $SO_2$ mixing ratio, separated into the southeast and southwest wind sectors for 2014 (A) and 2015 (B). $SO_2$ in the southwest sector showed diel variability that is largely consistent with DMS oxidation. $SO_2$ in the southeast sector was significantly lower and less variable in 2015 than in 2014.

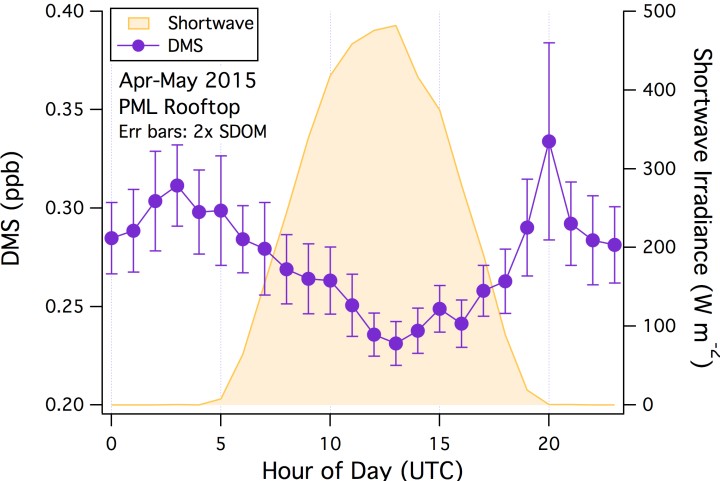

Figure 6. Average diel variability in DMS mixing ratio and shortwave irradiance, measured from the PML rooftop between April and May 2015 (wind from the marine sector). Error bars indicate two standard errors.

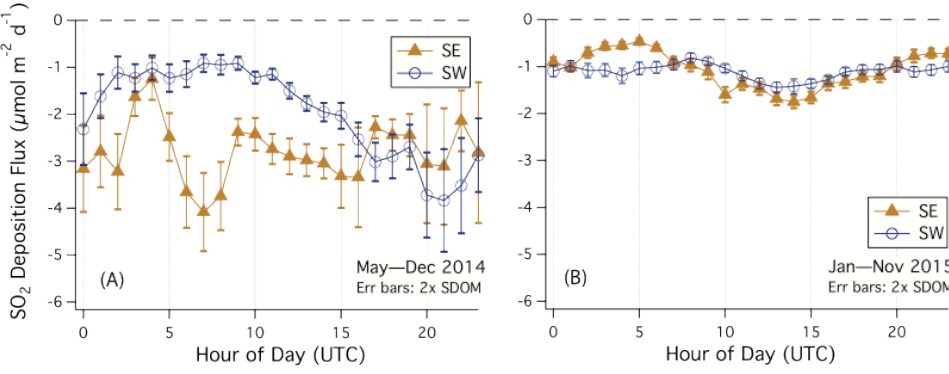

Figure 7. Mean diel cycles in the wind speed-dependent $SO_2$ deposition flux, separated into the southeast and southwest wind sectors for year 2014 (A) and 2015 (B). Error bars indicate two standard errors. $SO_2$ deposition flux was significantly greater in the southeast sector than in the southwest sector in 2014.




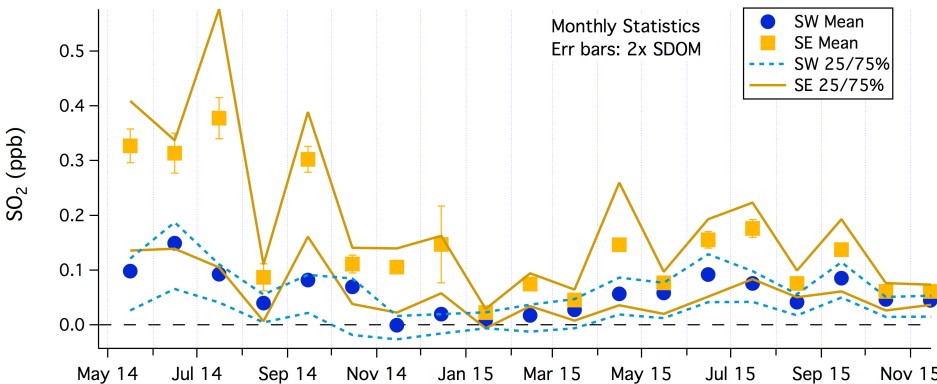

5    Figure 8. Monthly means and 25[th]/75[th] percentiles of $SO_2$ mixing ratio from southeast and southwest wind sectors. $SO_2$ from the southwest shows a clear seasonal cycle, with higher values in summer/early autumn, and lower values in winter/early spring. A similar underlying seasonal variability is also apparent in $SO_2$ from the southeast sector.

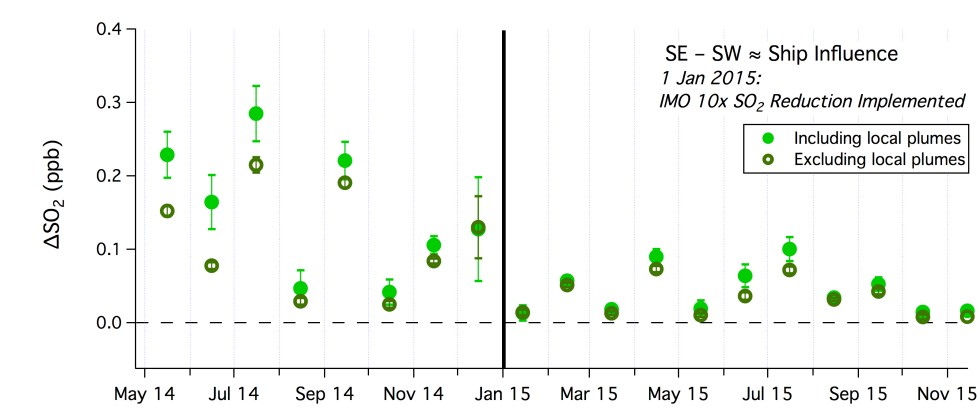

Figure 9. The difference between monthly averaged $SO_2$ mixing ratios from the southeast and southwest ($\Delta SO_2$), which we consider to approximately represent ship emissions. $\Delta SO_2$ including and excluding local ship plumes are shown. Error bars are propagated from 2 times the standard errors from each wind sector. Solid vertical line indicates the 1[st] Jan 2015 mandate for
15   reduction in ship $SO_2$ emissions.

20





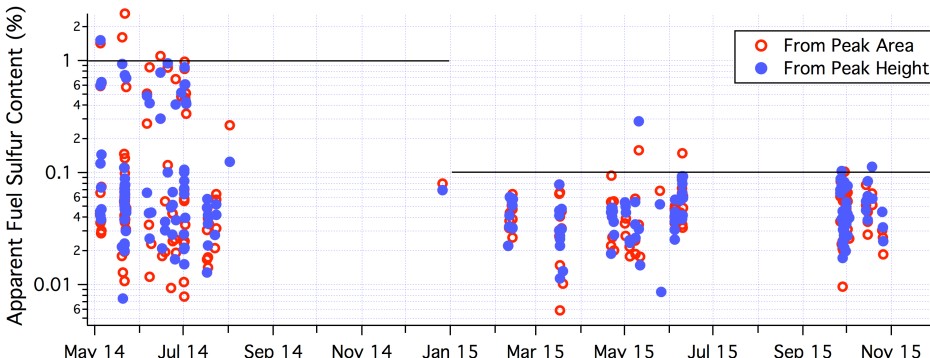

Figure 10. Apparent fuel sulfur content (FSC) estimated from peak areas as well as peak heights of coincidental $SO_2$ and $CO_2$ plumes (1-minute averaged data). Solid horizontal lines indicate IMO sulfur emission limits in European waters: 1% SFC in 2014, 0.1% SFC in 2015.

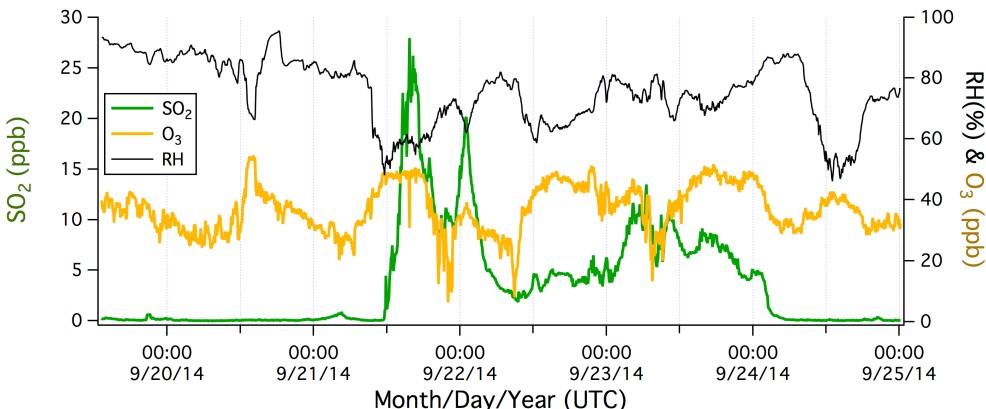

Figure A1. Very high $SO_2$ mixing ratios were observed in Icelandic volcano plumes. Elevated $SO_2$ sometimes coincided with
10    increased $O_3$ and reduced relative humidity (RH), consistent with entrainment from the free troposphere.