# Peer review of "Attribution of Atmospheric Sulfur Dioxide over the English Channel to Dimethylsulfide and Changing Ship Emissions"

_Atmospheric Chemistry and Physics, 2016_

## Referee Comment (RC1) · Anonymous Referee #1 · 4 Mar 2016

General comments

This is a well written and novel paper which uses observations from the Point Penlee Atmospheric Observatory to explore the importance of ship and biogenic sources to SO2 over the English Channel. The measurements reported are a valuable indicator of the effectiveness of Sulfur emission reduction schemes. The scientific questions explored are well within the scope of ACP and I recommend publication after the following minor recommendations are addressed.

Specific comments

Page 2 line 17 suggest 'source to' replaced with 'contributor to'

[Figure]

Line 20 " aerosols resulting from ship emissions contribute to tens of thousands of cases of premature mortality" Suggest insert 'cases of'

Page 4 line 2 "SO2 and O3 were blanked simultaneously"- blanked could be mistaken for capped off. Suggest 'blank measurements of SO2 and O3 were made simultaneously..."

Line 4 "We checked the calibration of the SO2 instrument occasionally" – be more specific about frequency – whether monthly, yearly etc

Page 4 line 8. SO2 measured at a height of 2m and CO2 at 18m. This is a significant height difference and I wonder might lead to de-coupling of air masses sampled at different heights? (particularly important for FSC calculations). Is there a measure of wind direction, speed at these two heights for comparison?

Page 4 Results A map of the Southern part of UK with the location of the observatory is needed to give perspective to where the observatory sits in relation to Plymouth Sound and the English Channel. This is provided in Yang et al 2016 but would also be useful here. Currently statements such as Page 4 line 24-25 "The wind sector between 110 and 250 deg is completely unobstructed by land" is not obvious looking at Fig 1.

Page 4 line 22 The wind rose shows 2 distinct dominant wind directions (SW and NNE). Suggest add a sentence describing land use/potential sources to the NNE.

More generally, the wind rose shows there is a low frequency of winds from the SE, with data from this direction an important focus of the paper. Could the authors describe how many hours of data they have used from the SE and SW directions respectively in their analyses?

Line 26 "Winds from between 50 and 110 deg face the eastern side of the Plymouth Sound, which is busy with ship traffic." Please provide an estimate of the volume of ship traffic for both the English Channel and Plymouth Sound.

Page 5, line 5 "The lowest SO2 mixing ratios were observed in the western, terrestriallyinfluenced wind sector in both years" It's not clear from Fig 1 that western direction is terrestrially influenced, hence need for a more regionally-scaled map

Page 5 line 13 "A lower, broader peak in $SO_2$ can also be observed between about 18:30 to 20:00 UTC." This $SO_2$ peak seems to correspond with a decrease in $O_3$, but not an increase in $CO_2$ which would be expected to be enhanced alongside $SO_2$. Please comment on this.

Line 19 "...as well as the busiest part of the shipping lanes." Could a figure be included which shows the shipping density in the SE versus the SW direction?

Page 6, line 14 "Recently-calibrated transmission efficiencies from the manufacturer (Ionicon, Austria) and kinetic reaction rates from Zhao and Zhang (2004) were used to derive the DMS mixing ratio." As DMS was not specifically calibrated during measurements, please provide an estimate of measurement uncertainty. Please comment on how this uncertainty impacts the diel amplitude of DMS and the calculated mixing ratio of $SO_2$ from oxidation of DMS.

Page 8 Line 29 In addition to the horizontal distribution of ship plumes, please comment on the likely vertical distribution of ship plumes observed at PPOA, which may be especially important given the different inlet heights of $SO_2$ and $CO_2$.

Page 9 line 6 "A ship 100 km away would have a plume that is observable for nearly an hour". Suggest add 'theoretically' to this sentence, as significant dilution of plume over 100km would make detecting enhancement of $SO_2$ and $CO_2$ very difficult?

Page 10 line 13 – how does the absolute FSC % from PPOA compare with recent estimates by Kattner and Beecken 2015?

Page 10 line 18 – please comment on how different inlet heights of $SO_2$ and $CO_2$ may add to uncertainty in estimating FSC

Line 22 "Long-term records of another tracer (e.g. nitrogen oxides or particle number)" Suggest that black carbon would be a better indicator for ship exhaust than particle

number, as particle number may be enhanced by local biogenically driven events.

Typing/technical errors Page 10, Line 20 A higher SO2' remove apostrophe after SO2

---

## Referee Comment (RC2) · Anonymous Referee #2 · 7 Mar 2016

This manuscript presents an important dataset that explores the contribution of shipping to atmospheric sulphur dioxide concentrations on the English Channel coast. I recommend that the authors consider revision in respect of the following points before publication.

1. The change in SECA regulations in January 2015 involved a 10-fold reduction in the maximum sulphur content of bunker fuel. However, the abstract (lines 14-16) states that a threefold reduction in $SO_2$ emissions was observed, and that there was a high level of SECA compliance in 2015. The differing factors of 3 and 10 are not discussed: the reason becomes clear from Figure 10, which shows that many ships were already complying with the 2015 regulations in 2014. This point should be clearly made in

order to avoid confusion (also in the Conclusion, p11, line 24).

2. p2, lines 24-25: needs rephrasing, e.g. "These regulations aim to reduce sulphur emission tenfold in SECAs by reducing the maximum allowed sulphur content of fuel from 1% to 0.1% by mass." The later sentence on open ocean regulations should also make clear that the percentages apply to the sulphur content of bunker fuel. It could be worth mentioning here that the SECA regulations allow ships to use scrubber technology as an alternative to low-sulphur fuel: the mention of scrubbers on p8, line 26 is otherwise mysterious to the uninitiated.

3. p5, line 20: in Figure 4, the sector with few mixing ratios over 0.5 ppb in 2015 is southwest, not southeast.

4. p5, line 29 – p6, line 1: Why not use the same months in both years for this averaging?

5. p7, line 9: Liss and Slater (1974) state that both the hydration and the subsequent oxidation of SO2 are rapid, but they gave no reference to the oxidation kinetics. It may well be that rapid hydration is enough to justify the assumption of near zero concentration of dissolved SO2. This point should be discussed more fully. Rate constants for the kinetics of sulphite oxidation can be found in Zhang and Millero (Geochim. Cosmochim. Acta, 57, 1705-1718, 1993).

6. p11, line 1: It is not clear what reductions are being discussed, in particular how does a reduction in 2014 arise?

---

## Author Comment (AC1) · 14 Mar 2016

Author Comment with regard to:

**"Attribution of Atmospheric Sulfur Dioxide over the English Channel to Dimethylsulfide and Changing Ship Emissions"**
by M. Yang et al.

14 March, 2016

Many thanks for the thoughtful *comments and suggestions from Anonymous Referee #1.* We are very glad to hear that the referee found our contribution valuable. Below are our replies to the specific comments, which are in *italic*.

*Anonymous Referee #1*
*General comments*
*This is a well written and novel paper which uses observations from the Point Penlee Atmospheric Observatory to explore the importance of ship and biogenic sources to SO2 over the English Channel. The measurements reported are a valuable indicator of the effectiveness of Sulfur emission reduction schemes. The scientific questions explored are well within the scope of ACP and I recommend publication after the following minor recommendations are addressed.*

*Specific comments*
*Page 2 line 17 suggest 'source to' replaced with 'contributor to'*
Suggestion accepted.

*Line 20 " aerosols resulting from ship emissions contribute to tens of thousands of cases of premature mortality" Suggest insert 'cases of'*
Suggestion accepted.

*Page 4 line 2 "SO2 and O3 were blanked simultaneously"- blanked could be mistaken for capped off. Suggest 'blank measurements of SO2 and O3 were made simultaneously."*
Suggestion accepted.

*Line 4 "We checked the calibration of the SO2 instrument occasionally" – be more specific about frequency – whether monthly, yearly etc*
We change the word occasionally to "twice a year."

*Page 4 line 8. SO2 measured at a height of 2m and CO2 at 18m. This is a significant height difference and I wonder might lead to de-coupling of air masses sampled at different heights? (particularly important for FSC calculations). Is there a measure of wind direction, speed at these two heights for comparison?*
Thanks for the comment. The ground level of the observatory is about 11 m above mean sea level. SO2 was measured at ~13 m above mean sea level (2 m above ground) while CO2 was generally measured at ~18 m above sea level. We did not have simultaneous wind measurements at these two heights. However, a significant difference in wind

direction between 13 and 18 m (both within the surface layer of the atmosphere) is generally not expected.

*Page 4 Results A map of the Southern part of UK with the location of the observatory is needed to give perspective to where the observatory sits in relation to Plymouth Sound and the English Channel. This is provided in Yang et al 2016 but would also be useful here. Currently statements such as Page 4 line 24-25 "The wind sector between 110 and 250 deg is completely unobstructed by land" is not obvious looking at Fig 1.*
Thanks for the comment. We've added the close up map below. The yellow circle indicates the location of the observatory. The Plymouth Sound spans ~4 km east of PPAO.

[Figure]

*Page 4 line 22 The wind rose shows 2 distinct dominant wind directions (SW and NNE). Suggest add a sentence describing land use/potential sources to the NNE.*
We will add that the NNE sector is likely influenced by emissions from terrestrial anthropogenic sources.

*More generally, the wind rose shows there is a low frequency of winds from the SE, with data from this direction an important focus of the paper. Could the authors describe how many hours of data they have used from the SE and SW directions respectively in their analyses?*
On average, winds came from our SW sector 7.5 days a month, and came from the SE sector 2.1 days a month.

*Line 26 "Winds from between 50 and 110 deg face the eastern side of the Plymouth Sound, which is busy with ship traffic." Please provide an estimate of the volume of ship traffic for both the English Channel and Plymouth Sound.*
According to the Devonport Naval Base Ship Movement Report, the total number of ships in the Plymouth Sound varies from about 4000 per month in winter to 6000 per month in summer. The volume of ship traffic in the English Channel is about 15000 per month (Maritime and Coastguard Agency, 2007).

*Page 5, line 5 "The lowest SO2 mixing ratios were observed in the western, terrestrially-*

*influenced wind sector in both years"* It's not clear from Fig 1 that western direction is *terrestrially influenced, hence need for a more regionally-scaled map*
See map above.

*Page 5 line 13 "A lower, broader peak in SO2 can also be observed between about 18:30 to 20:00 UTC." This SO2 peak seems to correspond with a decrease in O3, but not an increase in CO2 which would be expected to be enhanced alongside SO2. Please comment on this.*
Due to the high background mixing ratio in $CO_2$ (~400 ppm), ship plumes tend to result in much smaller (additional) signal:background ratios in $CO_2$ than in $SO_2$. As a result, $CO_2$ emitted from point sources tends to quickly become indistinguishable from the background with increasing distance (i.e. greater air dilution/dispersion). This is why we only focused on sharp, low plumes in the calculations of the fuel sulfur content.

*Line 19 ". . .as well as the busiest part of the shipping lanes." Could a figure be included which shows the shipping density in the SE versus the SW direction?*
We will refer to the ship's AIS (Automatic Identification System) maps shown by Jalkanen et al ACP 2016.

*Page 6, line 14 "Recently-calibrated transmission efficiencies from the manufacturer (Ionicon, Austria) and kinetic reaction rates from Zhao and Zhang (2004) were used to derive the DMS mixing ratio." As DMS was not specifically calibrated during measurements, please provide an estimate of measurement uncertainty. Please comment on how this uncertainty impacts the diel amplitude of DMS and the calculated mixing ratio of SO2 from oxidation of DMS.*
Without direct calibration, the uncertainty in the atmospheric DMS mixing ratio by the PTR-MS is ≤40%. A worst case ~40% lower atmospheric DMS would still be able to account for most of the observed $SO_2$ from the southwest sector.

*Page 8 Line 29 In addition to the horizontal distribution of ship plumes, please comment on the likely vertical distribution of ship plumes observed at PPOA, which may be especially important given the different inlet heights of SO2 and CO2.*
The difference in height between the $SO_2$ and $CO_2$ measurements was only 5 m (18-13 m). Eq. 2 from von Glasow et al (2003) predicts that just 10 seconds after emission, a ship plume will have already expanded vertically to a height of 20 m from the surface. It seems highly unlikely for our $SO_2$ and $CO_2$ sensors to be sampling different airmasses under typical meteorological conditions in the marine atmosphere.

*Page 9 line 6 "A ship 100 km away would have a plume that is observable for nearly an hour". Suggest add 'theoretically' to this sentence, as significant dilution of plume over 100km would make detecting enhancement of SO2 and CO2 very difficult?*
Agree. Suggestion accepted.

*Page 10 line 13 – how does the absolute FSC % from PPOA compare with recent estimates by Kattner and Beecken 2015?*

The mean FSC observed at PPAO (~0.17%) was about half of what was observed by Kattner et al (2015) for the year 2014. In 2015, mean FSC at PPAO (~0.047%) was fairly comparable to observations from Kattner et al (2015). Beecken et al. (2015) observed a bimodal distribution in FSC for year 2011 and 2012; the lower mode centered around ~0.25% and the higher mode centered around ~0.9%.

*Page 10 line 18 – please comment on how different inlet heights of SO2 and CO2 may add to uncertainty in estimating FSC*
We don't think the small height difference (5 m) significantly contributes to the uncertainty in FSC. However, using the same gas inlet for the SO2 and CO2 instruments and recording data from both on the same PC would help to better synchronize the two gas measurements.

*Line 22 "Long-term records of another tracer (e.g. nitrogen oxides or particle number)" Suggest that black carbon would be a better indicator for ship exhaust than particle number, as particle number may be enhanced by local biogenically driven events.*
Thanks for the suggestion.

*Typing/technical errors Page 10, Line 20 A higher SO2' remove apostrophe after SO2*
This is not an error. The apostrophe indicates deviation in SO2 mixing ratio from the background. See line 17 on p. 9.

---

## Author Comment (AC2) · 14 Mar 2016

Author Comment with regard to:

**"Attribution of Atmospheric Sulfur Dioxide over the English Channel to Dimethylsulfide and Changing Ship Emissions"**
by M. Yang et al.

14 March, 2016

Many thanks for the thoughtful *comments and suggestions from Anonymous Referee #2.* We are very glad to hear that the referee found our contribution valuable. Below are our replies to the specific comments, which are in *italic*.

**Anonymous Referee #2**

*This manuscript presents an important dataset that explores the contribution of shipping to atmospheric sulphur dioxide concentrations on the English Channel coast. I recommend that the authors consider revision in respect of the following points before publication.*

*1. The change in SECA regulations in January 2015 involved a 10-fold reduction in the maximum sulphur content of bunker fuel. However, the abstract (lines 14-16) states that a threefold reduction in SO2 emissions was observed, and that there was a high level of SECA compliance in 2015. The differing factors of 3 and 10 are not discussed: the reason becomes clear from Figure 10, which shows that many ships were already complying with the 2015 regulations in 2014. This point should be clearly made in order to avoid confusion (also in the Conclusion, p11, line 24).*
Thanks for the comment. We have clarified this in the revision.

*2. p2, lines 24-25: needs rephrasing, e.g. "These regulations aim to reduce sulphur emission tenfold in SECAs by reducing the maximum allowed sulphur content of fuel from 1% to 0.1% by mass." The later sentence on open ocean regulations should also make clear that the percentages apply to the sulphur content of bunker fuel. It could be worth mentioning here that the SECA regulations allow ships to use scrubber technology as an alternative to low-sulphur fuel: the mention of scrubbers on p8, line 26 is otherwise mysterious to the uninitiated.*
Suggestions accepted. Thanks.

*3. p5, line 20: in Figure 4, the sector with few mixing ratios over 0.5 ppb in 2015 is southwest, not southeast.*
There were indeed few occurrences >0.5 ppb in 2015 for the southwest sector. Though here we were making the point that the distribution shifted towards lower SO2 mixing ratios in the southeast.

*4. p5, line 29 – p6, line 1: Why not use the same months in both years for this averaging?*
We thought it would be useful to show the SO2 diel averages from nearly a whole year (2015). Seasonal variations are illustrated in Fig. 8 and 9.

*5. p7, line 9: Liss and Slater (1974) state that both the hydration and the subsequent oxidation of SO2 are rapid, but they gave no reference to the oxidation kinetics. It may well be that rapid hydration is enough to justify the assumption of near zero concentration of dissolved SO2. This point should be discussed more fully. Rate constants for the kinetics of sulphite oxidation can be found in Zhang and Millero (Geochim. Cosmochim. Acta, 57, 1705-1718, 1993).*

Thanks for the comment. As summarized by Schwartz (1992), SO2 dissociates almost instantaneously to form $HSO_3^-$ and then sulfite in seawater. The effective solubility of SO2 in seawater (pH ~8) due to this chemical enhancement is very large (dimensionless water:air solubility of about 5e8), which means that air-sea SO2 exchange should be gas phase controlled.

Oxidation to sulfate permanently removes sulfite from the ocean with a time scale of minutes to hours. This results in low sulfite concentration in the surface ocean (e.g. a few µM or less, Campanella et al 1995; Hayes et al 2006). Even a large dissolved SO2 concentration of 10 µM would only equate to an equilibrium atmospheric SO2 concentration of 2e-5 nmole/L of air (equivalent to 0.5 ppt, 2-3 orders of magnitude lower than typical atmospheric mixing ratio). From these calculations, we see that the air-sea concentration gradient in SO2 resides essentially all in the atmosphere.

*6. p11, line 1: It is not clear what reductions are being discussed, in particular how does a reduction in 2014 arise?*

Sorry for the confusion. "Reduction" here refers to the difference in the computed mean SO2 mixing ratio caused by excluding the very coastal ship plumes.

**References**:
Campanella, L., Cipriani, P., Martini, T.M., Sammartino, M.P., Tomassetti, M.: New enzyme sensor for sulfite analysis in sea and river water samples. Anal. Chim. Acta 305(1–3), 32–41, 1995.

Hayes, M., Taylor, G., Aster, Y.M., Scranton, M.J., Vertical distributions of thiosulfate and sulfite in the Cariaco Basin. Limnology and Oceanography 01/2006; 51(1):280-287. DOI: 10.4319/lo.2006.51.1.0280, 2006.

Schwartz, S.E.: Factors governing dry deposition of gases to surface water. In: Schwartz, S.E., Slinn, W.G.N. (eds.) Precipitation Scavenging and Atmosphere-Surface Exchange: Volume ii. Hemisphere Publishing Corp, Washington, 1992.

---

## Author Response (AR1)

[revised manuscript text omitted]

Mingxi Yang 3/24/16 10:28 AM
Mingxi Yang 3/24/16 10:28 AM
Mingxi Yang 3/24/16 10:28 AM
Mingxi Yang 3/24/16 10:28 AM
Mingxi Yang 3/24/16 10:28 AM
Mingxi Yang 3/24/16 10:28 AM
Mingxi Yang 3/24/16 10:28 AM
Mingxi Yang 3/24/16 10:28 AM
Mingxi Yang 3/24/16 10:28 AM
Mingxi Yang 3/24/16 10:28 AM
Mingxi Yang 3/24/16 10:28 AM
Mingxi Yang 3/24/16 10:28 AM
Mingxi Yang 3/24/16 10:28 AM
Mingxi Yang 3/24/16 10:28 AM
Mingxi Yang 3/24/16 10:28 AM
Mingxi Yang 3/24/16 10:28 AM
Mingxi Yang 3/24/16 10:28 AM
... [4]

---

## Editor Decision (ED1)

As a general note, there are a large number of changes in the revised manuscript which are **not** in response to the reviewer comments but rather seem to be 'proof reading' changes from the authors. In contrast, many of the reviewers comments were only addressed in the reviews and not in the article. The changes made without reference to reviewer comments were substantial and in some cases change information the reviewers referred to. Therefore, I have decided to send to revised article back to the reviewers to be sure that all their comments are addressed adequately and that they accept the changes made.

In future, the authors should ensure that they conduct proof reading **before** submission of the article. All reviewer comments should result in a change in the manuscript and not just a clarification in the author response – unless there is a clearly stated disagreement with the reviewer comment. Even if the reviewer has simply misunderstoof or missed a point, the article should be made more clear so that readers do not face the same problem.

**Response to comments from Reviewer 1:**

*More generally, the wind rose shows there is a low frequency of winds from the SE, with data from this direction an important focus of the paper. Could the authors describe how many hours of data they have used from the SE and SW directions respectively in their analyses?*

On average, winds came from our SW sector 7.5 days a month, and came from the SE sector 2.1 days a month.

- Please add this information to the revised manuscript.

*Line 26 "Winds from between 50 and 110 deg face the eastern side of the Plymouth Sound, which is busy with ship traffic." Please provide an estimate of the volume of ship traffic for both the English Channel and Plymouth Sound.*

According to the Devonport Naval Base Ship Movement Report, the total number of ships in the Plymouth Sound varies from about 4000 per month in winter to 6000 per month in summer. The volume of ship traffic in the English Channel is about 15000 per month (Maritime and Coastguard Agency, 2007).

- Please add this information to the revised manuscript.

*Page 6, line 14 "Recently-calibrated transmission efficiencies from the manufacturer (Ionicon, Austria) and kinetic reaction rates from Zhao and Zhang (2004) were used to derive the DMS mixing ratio." As DMS was not specifically calibrated during measurements, please provide an estimate of measurement uncertainty. Please comment on how this uncertainty impacts the diel amplitude of DMS and the calculated mixing ratio of $SO_2$ from oxidation of DMS.*

Without direct calibration, the uncertainty in the atmospheric DMS mixing ratio by the PTR-MS is ≤40%. A worst case ~40% lower atmospheric DMS would still be able to account for most of the observed SO2 from the southwest sector.

- Please add this information to the revised manuscript.

*Page 8 Line 29 In addition to the horizontal distribution of ship plumes, please comment on the likely vertical distribution of ship plumes observed at PPOA, which may be especially important given the different inlet heights of SO2 and CO2.*

The difference in height between the SO2 and CO2 measurements was only 5 m (18-13 m). Eq. 2 from von Glasow et al (2003) predicts that just 10 seconds after emission, a ship plume will have already expanded vertically to a height of 20 m from the surface. It seems highly unlikely for our SO2 and CO2 sensors to be sampling different airmasses under typical meteorological conditions in the marine atmosphere.

- Please add this information to the revised manuscript. This reviewer asks several times about the different inlet heights: please add more information to the paper to address the multiple concerns of the reviewer and anticipate similar questions in readers.

*Page 10 line 13 – how does the absolute FSC % from PPOA compare with recent estimates by Kattner and Beecken 2015?*

The mean FSC observed at PPAO (~0.17%) was about half of what was observed by Kattner et al (2015) for the year 2014. In 2015, mean FSC at PPAO (~0.047%) was fairly comparable to observations from Kattner et al (2015). Beecken et al. (2015) observed a bimodal distribution in FSC for year 2011 and 2012; the lower mode centered around ~0.25% and the higher mode centered around ~0.9%.

- Please add this information to the revised manuscript.

***Response to comments from Reviewer 2:***

*3. p5, line 20: in Figure 4, the sector with few mixing ratios over 0.5 ppb in 2015 is southwest, not southeast.*

There were indeed few occurrences >0.5 ppb in 2015 for the southwest sector. Though here we were making the point that the distribution shifted towards lower SO2 mixing ratios in the southeast.

- Please clarify this point in the revised article.

*5. p7, line 9: Liss and Slater (1974) state that both the hydration and the subsequent oxidation of SO2 are rapid, but they gave no reference to the oxidation kinetics. It may well be that rapid hydration is enough to justify the assumption of near zero concentration of dissolved SO2. This point should be discussed more fully. Rate constants for the kinetics of sulphite oxidation can be found in Zhang and Millero (Geochim. Cosmochim. Acta, 57, 1705-1718, 1993).*

Thanks for the comment. As summarized by Schwartz (1992), SO2 dissociates almost instantaneously to form HSO3 - and then sulfite in seawater. The effective solubility of SO2 in seawater (pH ~8) due to this chemical enhancement is very large (dimensionless water:air solubility of about 5e8), which means that air-sea SO2 exchange should be gas phase controlled. Oxidation to sulfate permanently removes sulfite from the ocean with a time scale of minutes to hours. This results in low sulfite concentration in the surface ocean (e.g. a few μM or less, Campanella et al 1995; Hayes et al 2006). Even a large dissolved SO2 concentration of 10 μM would only equate to an equilibrium atmospheric SO2 concentration of 2e-5 nmole/L of air (equivalent to 0.5 ppt, 2-3 orders of magnitude lower than typical atmospheric mixing ratio). From these calculations, we see that the airsea concentration gradient in SO2 resides essentially all in the atmosphere.

- Clarify this response in detail in the revised article.

*6. p11, line 1: It is not clear what reductions are being discussed, in particular how does a reduction in 2014 arise?*

Sorry for the confusion. "Reduction" here refers to the difference in the computed mean SO2 mixing ratio caused by excluding the very coastal ship plumes.

- Please clarify this in the revised article.

**Additional comments:**

- P1 L29: Change oxidations back to oxidation